# Distinct Antibody Responses to Endemic Coronaviruses Pre- and Post-SARS-CoV-2 Infection in Kenyan Infants and Mothers

**DOI:** 10.3390/v14071517

**Published:** 2022-07-12

**Authors:** Caitlin I. Stoddard, Kevin Sung, Ednah Ojee, Judith Adhiambo, Emily R. Begnel, Jennifer Slyker, Soren Gantt, Frederick A. Matsen, John Kinuthia, Dalton Wamalwa, Julie Overbaugh, Dara A. Lehman

**Affiliations:** 1Human Biology Division, Fred Hutchinson Cancer Research Center, Seattle, WA 98109, USA; cstoddar@fredhutch.org; 2Public Health Sciences Division, Fred Hutchinson Cancer Research Center, Seattle, WA 98109, USA; ksung2@fredhutch.org (K.S.); matsen@fredhutch.org (F.A.M.IV); 3Department of Pediatrics and Child Health, University of Nairobi, Nairobi 00100, Kenya; eojeg@uonbi.ac.ke (E.O.); adhiamboj69@yahoo.com (J.A.); dalton.wamalwa@uonbi.ac.ke (D.W.); 4Department of Global Health, University of Washington, Seattle, WA 98195, USA; erb29@uw.edu (E.R.B.); jslyker@uw.edu (J.S.); kinuthia@uw.edu (J.K.); 5Department of Epidemiology, University of Washington, Seattle, WA 98195, USA; 6Département de Microbiologie, Infectiologie et Immunologie, Université de Montréal, Montréal, QC H3T 1J4, Canada; soren.gantt.med@ssss.gouv.qc.ca; 7Centre Hospitalier Universitaire Sainte-Justine, Montréal, QC H3T 1C5, Canada; 8Howard Hughes Medical Institute, Chevy Chase, MD 20815, USA; 9Department of Research and Programs, Kenyatta National Hospital, Nairobi 00202, Kenya

**Keywords:** SARS-CoV-2, endemic, infants, mothers, Kenya, cross-reactive, boosting, antibody, IgG, coronavirus

## Abstract

Pre-existing antibodies that bind endemic human coronaviruses (eHCoVs) can cross-react with SARS-CoV-2, which is the betacoronavirus that causes COVID-19, but whether these responses influence SARS-CoV-2 infection is still under investigation and is particularly understudied in infants. In this study, we measured eHCoV and SARS-CoV-1 IgG antibody titers before and after SARS-CoV-2 seroconversion in a cohort of Kenyan women and their infants. Pre-existing eHCoV antibody binding titers were not consistently associated with SARS-CoV-2 seroconversion in infants or mothers; however, we observed a very modest association between pre-existing HCoV-229E antibody levels and a lack of SARS-CoV-2 seroconversion in the infants. After seroconversion to SARS-CoV-2, antibody binding titers to the endemic betacoronaviruses HCoV-OC43 and HCoV-HKU1, and the highly pathogenic betacoronavirus SARS-CoV-1, but not the endemic alphacoronaviruses HCoV-229E and HCoV-NL63, increased in the mothers. However, eHCoV antibody levels did not increase following SARS-CoV-2 seroconversion in the infants, suggesting the increase seen in the mothers was not simply due to cross-reactivity to naively generated SARS-CoV-2 antibodies. In contrast, the levels of antibodies that could bind SARS-CoV-1 increased after SARS-CoV-2 seroconversion in both the mothers and infants, both of whom were unlikely to have had a prior SARS-CoV-1 infection, supporting prior findings that SARS-CoV-2 responses cross-react with SARS-CoV-1. In summary, we found evidence of increased eHCoV antibody levels following SARS-CoV-2 seroconversion in the mothers but not the infants, suggesting eHCoV responses can be boosted by SARS-CoV-2 infection when a prior memory response has been established, and that pre-existing cross-reactive antibodies are not strongly associated with SARS-CoV-2 infection risk in mothers or infants.

## 1. Introduction

The SARS-CoV-2 pandemic has caused a global catastrophe and is characterized by varying infection risks and clinical outcomes in those that become infected. Younger age groups have been associated with a lower likelihood of infection in numerous studies [1,2]. Several explanations for this phenomenon have been hypothesized, including the influence of cross-reactive immune responses to endemic human coronaviruses (eHCoVs), also known as seasonal or common colds which cause human coronaviruses. Many studies have shown that eHCoV antibody levels are increased upon SARS-CoV-2 infection [3,4,5,6,7,8,9,10,11], which may indicate “boosted” pre-existing memory responses that are cross-reactive. It remains unclear whether such cross-reactive antibody responses could modulate SARS-CoV-2 infection risk.

Additionally, while several studies have examined eHCoV antibody responses in children and adults [10,12,13,14,15,16], studies testing for eHCoV antibody responses in newborns or infants and studies that directly compare infants and adults are lacking. Infants are born with passively transferred eHCoV antibodies from their mothers that wane during the early months of life. Those less than 6 months of age are less likely to experience eHCoV infection compared to older children [17,18] and thus may not have memory responses that can be further stimulated by another HCoV infection. In addition, when infants are infected, their antibody responses may differ from those of adults [19,20], further underscoring the importance of studying eHCoV and SARS-CoV-2 antibody dynamics in infant populations.

Here, we profiled eHCoV antibodies in the infants and mothers by measuring the IgG titers to the spike protein of four eHCoVs, including two from the same genus as SARS-CoV-2 (betacoronaviruses HCoV-OC43 and HCoV-HKU1) and two alphacoronaviruses (HCoV-229E and HCoV-NL63) (Appendix A). We also measured the antibodies to the SARS-CoV-1 spike protein, which shares the most sequence homology with SARS-CoV-2 among the coronaviruses we included (76% identity, [21]; Appendix A). We leveraged a longitudinal cohort study of mothers and infants that did or did not seroconvert to SARS-CoV-2 to firstly test for differences in the eHCoV antibody titers between infants and mothers in naive and SARS-CoV-2-seroconverted samples, and then evaluate associations between pre-existing eHCoV titers and SARS-CoV-2 seroconversion during the study period.

## 2. Materials and Methods

### 2.1. Study Participants

A subset of mothers and infants in Nairobi, Kenya that were already enrolled in the Linda Kizazi Study, a prospective cohort study of mother-to-child virome transmission, consented to SARS-CoV-2 serology testing as previously described [22]. Mother–infant pairs attended clinic visits approximately every 3 months, at which time clinical data were collected, including recent diagnoses and healthcare visits, symptoms of illness at the time of the visit or since the last visit, and the history of current or recent medications or immunizations. Physical examinations were conducted at each clinic visit, and samples, including blood, were collected. The Kenyatta National Hospital-University of Nairobi Ethics and Research Committee and the University of Washington and Fred Hutchinson Institutional Review Boards approved of all human subject study procedures.

### 2.2. Sample Selection

Plasma samples collected between April 2019 and December 2020 were selected for this sub-study based on previous SARS-CoV-2 serology testing [22]. Notably, this study period was prior to the initiation of vaccination campaigns in Kenya and spans a timeframe in which the B.1 lineage of SARS-CoV-2, which includes the D614G mutation, was found to be globally predominant [22]. For the participants who seroconverted during the study period, up to 3 longitudinal plasma samples were included: the “first seropositive” was the plasma sample in which SARS-CoV-2 antibodies were first detected by ELISA testing [22], the “last negative” was the most recent plasma sample collected prior to the first seropositive sample time point, and a “pre-pandemic” sample (if available) was the most recent sample collected prior to October 2019 to ensure no possible exposure to SARS-CoV-2. Samples from participants that did not seroconvert to SARS-CoV-2 during the study period include up to 2 longitudinal plasma samples: a “time-matched seronegative” sample, which was collected during the time window of the last negative sample from the seroconverters (December 2019–April 2020); and a “pre-pandemic” sample as described above.

### 2.3. Multiplexed Chemiluminescent Antibody Binding Assay with Plasma

The plasma samples were heat-inactivated for 60 min at 56 °C prior to 1:5000 dilution. eHCoV antibody levels were determined using Mesoscale Diagnostic’s V-PLEX Coronavirus Panel 2 which includes pre-fusion stabilized spike trimers from all four eHCoVs, plus SARS-CoV-1 spotted together in individual wells in a 96-well format. Binding specificity for HCoV-OC43 and SARS-CoV-2 spike has been previously determined for this commercially available assay [23,24]. The assay was performed following the manufacturer’s instructions. Diluted samples, along with manufacturer-provided calibrators and controls, were applied to blocked plates and incubated for 2 h. Washed plates were incubated with detection antibody for 1 h, followed by the addition of MSD GOLD Read Buffer B. The plates were read on the MSD instrument and the raw data were processed in MSD Discovery Workbench software (version 4.0) (Mesoscale Diagnostics, Rockville, MO, USA). IgG antibody levels for each antigen were calculated in Workbench based on the calibrator standard curve fit and reported in Arbitrary Units/mL (AU/mL). IgG concentrations that were below the manufacturer’s detection or curve fit limits were set to 0.

### 2.4. Statistical Analyses

Wilcoxon rank-sum or Wilcoxon signed-rank tests were performed to compare the samples that were unmatched or matched, respectively. *p*-values were adjusted for multiple hypothesis testing by applying Holm–Bonferroni correction. All statistical tests were performed using SciPy [25] and statsmodels [26] software tools.

## 3. Results

### 3.1. Participant Groups and Sample Timing

Longitudinal plasma samples collected from an ongoing study of mother-to-child virome transmission in Nairobi, Kenya (the Linda Kizazi cohort) were previously tested for SARS-CoV-2 nucleocapsid seroconversion by enzyme-linked immunosorbent assay (ELISA) [22]. The mothers and infants were grouped as either seroconverters or never-seropositive for SARS-CoV-2 during the follow-up and included in this sub-study (from April 2019 to December 2020; Figure 1A,B and Appendix A). The plasma samples from the seroconverters (*n* = 50) included pre-pandemic (as available prior to October 2019; mothers, *n* = 14; infants, *n* = 5), last seronegative (mothers, *n* = 35; infants, *n* = 11), and first seropositive samples (mothers, *n* = 36; infants, *n* = 14). For the individuals that never seroconverted in the study period (*n* = 121), we selected a pre-pandemic sample (when available; mothers *n* = 21; infants, *n* = 10), as well as a pandemic-era sample, termed “time matched seronegative”, that overlapped the time period (from December 2019 to April 2020) of the last negative samples from the seroconverters (mothers, *n* = 62; infants, *n* = 56; Figure 1A,B and Appendix A). None of the mothers seroconverted during pregnancy, so any detectable SARS-CoV-2 antibodies in the infant plasma used to determine serostatus [22] were a result of postnatal infection and not due to passive transfer of SARS-CoV-2 antibodies in utero. The median (IQR) infant age in the sample groups was as follows: pre-pandemic, 9.7 (6.7–10.6) weeks; last negative or time-matched negative, 25.1 (10.0–38.8) weeks; and first seropositive, 47.4 (33.0–65.7) weeks. In the Linda Kizazi cohort, approximately 20% of the infants and mothers reported one or more mild-to-moderate symptoms of COVID-19 at their first seropositive visit or since their last seronegative visit, and there were no reported hospitalizations or deaths due to COVID-19 in the cohort [22].

### 3.2. Longitudinal eHCoV Antibody Responses in SARS-CoV-2 Seroconverting and Non-Seroconverting Mothers and Infants

To test for the presence of IgG antibodies targeting eHCoVs, we compared antibody binding titers to spike from the four commonly circulating eHCoVs in the longitudinal plasma samples from the infants and mothers using a commercially available multiplexed chemiluminescent antibody binding immunoassay. In the pre-pandemic samples collected prior to the emergence of SARS-CoV-2, the mothers and infants displayed similar levels of antibodies against all four eHCoVs, as would be expected for systemic maternally transferred antibodies present in the plasma of infants at the age sampled, which was a median of 9.7 weeks of age (Figure 2, left column). In the pandemic-era samples collected most recently before SARS-CoV-2 seroconversion or in the time-matched window for non-seroconverting individuals, the mothers had significantly higher levels of antibodies targeting the four eHCoVs compared to the infants, which were a median of 25.1 weeks of age with the most pronounced difference for HCoV-NL63 (Figure 2, middle column). This difference was largely driven by lower median levels of infant antibodies, rather than an increase in systemic maternal antibodies, suggesting a waning of the passively transferred response in the infants over time. Similarly, upon seroconversion to SARS-CoV-2, the mothers exhibited significantly higher levels of eHCoV antibodies than the infants (Figure 2, right column). These results demonstrate differences in the infant and maternal antibody responses to eHCoVs in this cohort likely reflecting the more limited opportunity for eHCoV exposure in the early months of an infant’s life in part due to passive antibody protection.

### 3.3. SARS-CoV-2 Infection Is Associated with Increases in Betacoronavirus eHCoV Antibody Response

Our observation of significantly higher levels of eHCoV antibodies in the mothers versus the infants after SARS-CoV-2 seroconversion prompted us to test whether antibody levels increased between the last seronegative and first seropositive samples in the infants and mothers. To test whether SARS-CoV-2 infection was associated with increases in antibodies that bind to eHCoV in infants and mothers, we compared antibody levels longitudinally in the SARS-CoV-2 seroconverters. Between the last negative and first seropositive plasma samples, the antibody levels for both endemic betacoronaviruses, HCoV-OC43 and HCoV-HKU1, increased significantly in the mothers, but not the infants, suggesting a cross-reactive response that could be influenced by pre-existing eHCoV antibodies in adults (Figure 3A,B and Appendix A). Alphacoronavirus antibody levels did not increase in either group. Interestingly, antibody levels against the highly pathogenic betacoronavirus, SARS-CoV-1, increased most significantly between the last negative and first seropositive samples, and this was true for both the mothers and infants (Figure 3A,B and Appendix A). Given the lack of evidence for SARS-CoV-1 circulation in Kenya [27], this result suggests a prior exposure to the virus is not driving this increase, rather it reflects de novo responses to SARS-CoV-2 infection that recognize SARS-CoV-1, which shares a high degree of sequence homology.

### 3.4. Pre-Existing eHCoV Antibody Levels Are Not Strongly Associated with SARS-CoV-2 Serostatus

To test whether recent eHCoV antibody levels were associated with SARS-CoV-2 seroconversion, we compared eHCoV antibody binding titers between never seropositive and seroconverting infants and mothers in the last negative and time-matched seronegative samples. While we did not observe a relationship between prior eHCoV antibody titers and SARS-CoV-2 infection in the mothers, we observed a modest association between the HCoV-229E antibody binding titer and SARS-CoV-2 seronegativity in the infants; however, this result fell below statistical significance after correction for multiple hypothesis testing (Figure 4A,B). Similarly, we did not observe a statistically significant relationship between pre-pandemic eHCoV antibody binding titers and SARS-CoV-2 seroconversion in the infants or mothers (Appendix A); however, the number of samples in this group was more limited.

## 4. Discussion

The role of cross-reactive eHCoV antibody responses in SARS-CoV-2 infection or protection remains unclear, and there is a scarcity of data on this relationship in the earliest months of life. In this study, we measured eHCoV IgG antibody binding responses in infants and mothers just prior to and after SARS-CoV-2 seroconversion. We found higher levels of eHCoV antibody binding in the mothers versus the infants just prior to SARS-CoV-2 infection, which likely reflects the decreased probability of exposure to eHCoVs during the shorter infant lifespan.

Increased eHCoV antibody levels upon SARS-CoV-2 seroconversion have been observed in some cohorts [3,4,5,6,7,8,9,10] but not others [28,29], and when detected, they have been hypothesized to reflect a boosting of pre-existing cross-reactive responses. We observed significant increases in betacoronavirus but not alphacoronavirus antibody levels (SARS-CoV-1 in the infants and mothers; HCoV-OC43 and HCoV-HKU1 in the mothers) upon SARS-CoV-2 seroconversion compared to the most recent seronegative sample, suggesting that there are cross-reactive antibodies to SARS-CoV-2, which may be more likely to be present in the context of more closely related eHCoVs. Interestingly, we did not observe increased eHCoV antibody levels upon SARS-CoV-2 seroconversion in the infants, which likely reflects the absence of pre-existing eHCoV memory responses in infants due to both passive antibody protection and a limited period of exposure. We identified SARS-CoV-1 cross-reactive antibodies in both the infants and mothers, which are responses that are unlikely to reflect pre-existing memory responses given the lack of circulating SARS-CoV-1. Such cross-reactive responses suggest SARS-CoV-2 infection may induce naively generated cross-reactive responses that recognize SARS-CoV-1; however, the above analyses did not model for time due to confounding factors, including time since infection, the mothers’ postpartum date, and changing maternal antibody levels in the infants. Additional studies will be required to evaluate the influence of SARS-CoV-2 evolution and vaccination campaigns on cross-reactive responses to SARS-CoV-1 and eHCoVs in women and infants.

Pre-existing immune cross-protection and a lower median age have been hypothesized as correlates of protection against severe COVID-19 in sub-Saharan Africa [30,31]. However, whether pre-existing eHCoV antibodies are protective or increase the risk of SARS-CoV-2 infection remains controversial [32], and this relationship is particularly understudied in infants. In our study, prior eHCoV antibody titers were not strongly associated with SARS-CoV-2 seroconversion. Further study with larger cohorts will be needed to evaluate this association as our study is limited by sample size and potential heterogeneity in SARS-CoV-2 exposure risk in the study population. Together, these results demonstrate differences in eHCoV antibody responses pre- and post-SARS-CoV-2 infection between infants and mothers in Kenya, including evidence for HCoV-OC43 and HCoV-HKU1 antibody boosting upon SARS-CoV-2 seroconversion in mothers but not infants and provide a basis for further evaluation of cross-reactive eHCoV antibody responses in newborns and young infants in the context of SARS-CoV-2.

## Figures and Tables

**Figure 1 viruses-14-01517-f001:**
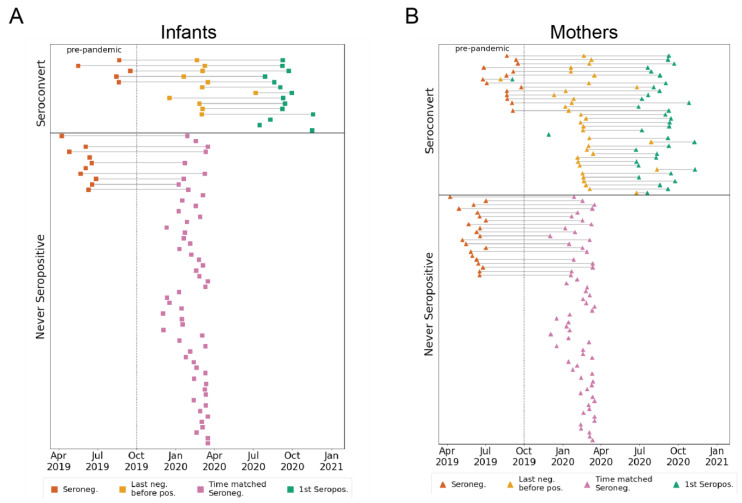
Participant groups and plasma sample timing. (**A**) Infants or (**B**) mothers were grouped based on nucleocapsid ELISA results [22] as either seroconverting or never seropositive in the sampling window from April 2019 to December 2020. Samples from seroconverters were selected as pre-pandemic (red), last seronegative (yellow), and first seropositive (green). For never seropositive individuals, pre-pandemic (red), and pandemic-era samples (pink) that overlap the calendar time window of the last seronegative samples in the seroconverting group, were selected.

**Figure 2 viruses-14-01517-f002:**
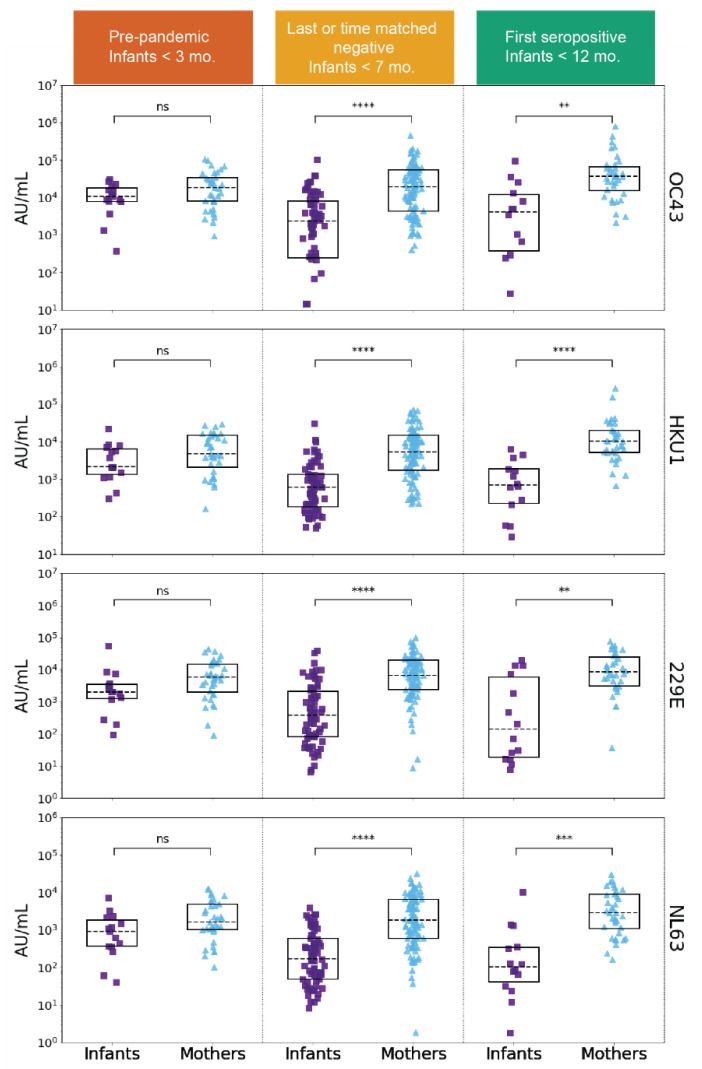
eHCoV IgG titers in SARS-CoV-2 naive and SARS-CoV-2 seroconverted plasma from infants and mothers. HCoV-OC43, HCoV-HKU1, HCoV-229E, and HCoV-NL63 spike IgG levels (AU/mL) in pre-pandemic (SARS-CoV-2 naive) plasma (left column) from never SARS-CoV-2 seropositive and eventually seroconverting infants (purple, *n* = 15) and mothers (blue, *n* = 35), last negative before SARS-CoV-2 seropositive or time-matched never seropositive infants (*n* = 67) and mothers (*n* = 97) (middle column), and first SARS-CoV-2 seropositive samples from infants (*n* = 14) and mothers (*n* = 36) (right column). The sample groups and median infant age are indicated in the colored headings. *p*-values (Wilcoxon rank-sum test) are corrected for multiple hypothesis testing (Holm–Bonferroni). (ns) *p* > 0.05, (**) *p* ≤ 0.01, (***) *p* ≤ 0.001, and (****) *p* ≤ 0.0001.

**Figure 3 viruses-14-01517-f003:**
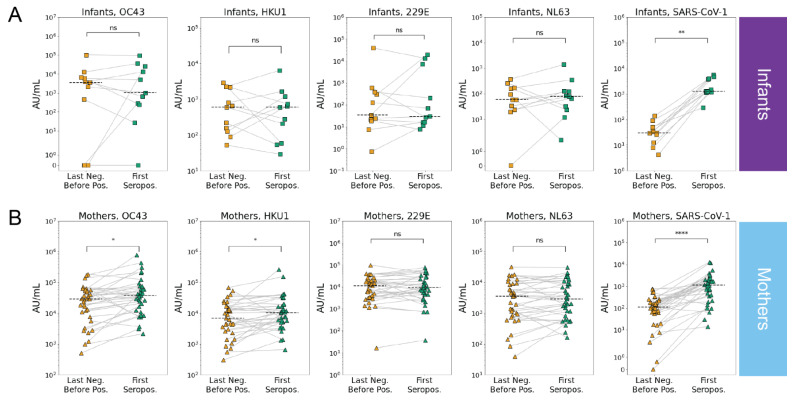
eHCoV and SARS-CoV-1 antibody titers immediately prior to and after SARS-CoV-2 seroconversion in infants and mothers. Last negative (yellow) and first seropositive (green) eHCoV spike IgG titers (AU/mL) in (**A**) infants (*n* = 11) and (**B**) mothers (*n* = 35). *p*-values (Wilcoxon signed rank test) are indicated and corrected for multiple hypothesis testing (Holm–Bonferroni). Significant comparisons (*p* < 0.05) are further indicated with an asterisk. (ns) *p* > 0.05, (*) *p* ≤ 0.05, (**) *p* ≤ 0.01, and (****) *p* ≤ 0.0001.

**Figure 4 viruses-14-01517-f004:**
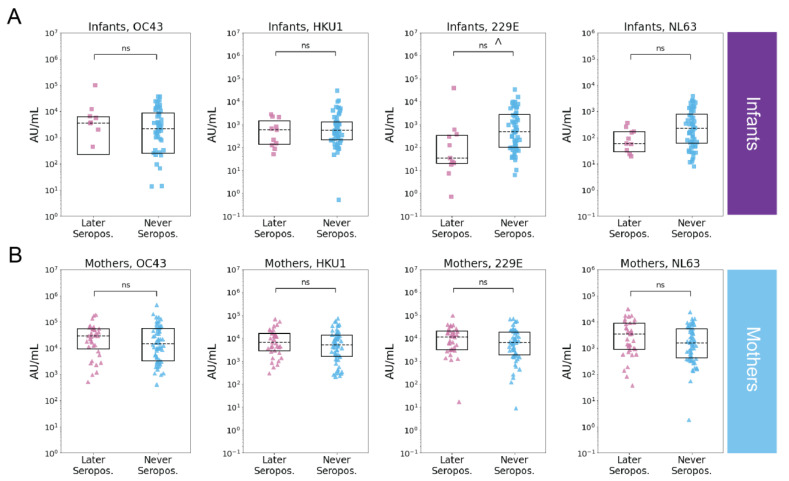
Relationship between last negative samples and SARS-CoV-2 serostatus in infants and mothers. Recent prior eHCoV spike IgG levels (AU/mL) in individuals that were later SARS-CoV-2 seropositive (pink) or seronegative (blue) in (**A**) infants (later seropositive *n* = 11, never seropositive *n* = 56) and (**B**) mothers (later seropositive *n* = 35, never seropositive *n* = 62). *p*-values (Wilcoxon rank-sum test) are indicated and corrected for multiple hypothesis testing (Holm–Bonferroni). (ns) *p* > 0.05 and *p* ≤ 0.05 prior to Holm–Bonferroni correction indicated with ^.

## Data Availability

Not applicable.

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
