# Peer review of "Distinct Antibody Responses to Endemic Coronaviruses Pre- and Post-SARS-CoV-2 Infection in Kenyan Infants and Mothers"

_viruses, 2022, doi:10.3390/v14071517_

Round 1

Reviewer 1 Report

The manuscript needs to be more explicit about waht antigens were used for serology data. And one key issue about the serology data is that implicit assumptions were made about high specificity towards each coronavirus antigen, which should be established first. Delineating against specific antigens might further untangle underlying correlations.

Reviewer 2 Report

The authors report the findings of their study on the antibody responses to endemic coronaviruses pre- and post-SARS-CoV-2 infection in a population fo infants and mothers from Kenya.

The approach is clearly explained and the results are well described.

Major points to be addressed:

A description of the cohort would be helpful (Table?). Especially since the study that keeps getting referred to is not available. Lines 75, 85, 88, 129, 136: "Begnel et al., in revision" is not good enough as a reference. Either submit that manuscript as a pre-print and reference the pre-print or indicate as data not shown.

Is there any documentation about the feeding mode for the infants ?(breastfeeding throughout the study?  stopped at a given age?) For "older" infants that are no longer being breastfed, the amount of detectable antibodies would be expected to be much lower than infant still being breastfed, contributing to the "waning of the passively transferred response". 

Is the sequence of the infecting variant known for all these infections? If not, is it possible to make assumptions regarding the circulating variants in the area at the time of the study (possibly D614G? )? Given that the sequence of SARS-CoV-2 Spike has evolved during the pandemic, the homology with SARS-CoV-1 spike would be expected to also change. Could the authors expand the discussion of their results to other variants? and what are the expected effects of vaccination on the findings?

Please expand the limitations of the study beyond lines 246-248, including circulating variants (see above).

Minor points:

line 52-53: "several studies" but only 1 reference listed.

Figure 3: it seems that some lines are cut off at the x-axis. Please indicate a limit of detection for the assay or adjust the y axis so that all the data points are shown.

line 261: orphaned title "5. Conclusions"

Is there any evidence that the antibodies binding SARS-CoV-1 are indeed also recognizing SARS-CoV-2 Spike? While understanding that the infant/mother samples are precious and might be scarce, is it possible to validate the cross reactivity (SARS-CoV1/2) in sera from another cohort by showing loss of binding if the sera is pre-incubated with SARS-CoV-2 spike and then tested for ELISA against SARS-CoV-1 Spike [or vice-versa]?

It would be helpful to add to the Methods section that all samples were collected before the vaccination campaign started in Kenya.

Reviewer 3 Report

This article talks about the pre-existing eHCoV antibody responses between mothers and infants prior to and after COVID seroconversion. The topic of this paper is very essential and has a significant impact to public health. Researchers have delineated very important findings of the cross-reactive eHCoV antibody responses, and have contributed to the field, especially from the aspects of newborns, since there’s not much work about this topic has been done. Overall, the paper is very well written and organized.

One suggestion: I would suggest the authors to add a table or a figure describing the relationships of different kinds of coronaviruses mentioned in the paper. 
